# TCBench: A Benchmark for Tropical Cyclone Track and Intensity Forecasting at the Global Scale

## Abstract

TCBench is a benchmark for evaluating global, short to medium-range (1-5 days) forecasts of tropical cyclone (TC) track and intensity. To allow a fair and model-agnostic comparison, TCBench builds on the IBTrACS observational dataset and formulates TC forecasting as predicting the time evolution of an existing tropical system conditioned on its initial position and intensity. TCBench includes state-of-the-art dynamical (TIGGE) and neural weather models (AIFS, Pangu-Weather, FourCastNet v2, GenCast). If not readily available, baseline tracks are consistently derived from model outputs using the TempestExtremes library. For evaluation, TCBench provides deterministic and probabilistic storm-following metrics. On 2023 test cases, neural weather models skillfully forecast TC tracks, while skillful intensity forecasts require additional steps such as post-processing. Designed for accessibility, TCBench helps AI practitioners tackle domain-relevant TC challenges and equips tropical meteorologists with data-driven tools and workflows to improve prediction and TC process understanding. By lowering barriers to reproducible, process-aware evaluation of extreme events, TCBench aims to democratize data-driven TC forecasting.

## 1 Introduction

Tropical cyclones (TCs), also called "hurricanes" or "typhoons" depending on the basin (American Meteorological Society, 2020), are globally devastating weather systems. In the U.S. alone, TCs caused over \$1.5 trillion in damages and over 7,000 deaths between 1980 and 2024 (NOAA National Centers for Environmental Information, 2025). With over half of the global population projected to live in the tropics and low-elevation coastal zones by 2050-2060 (Gu et al., 2021; Neumann et al., 2015), **improving TC forecasting is urgent** for risk mitigation and community resilience.

TC forecasting can be framed as the task of forecasting **two continuous time-series**:

**Track**: the time-series of storm locations in latitude $\phi \in [-50, 50]°$ and longitude $\lambda \in [0, 360]°$;

**Intensity**: commonly characterized by maximum sustained wind speed $V_{\max} \in \mathbb{R}_+$ (m s$^{-1}$) and minimum sea-level pressure $p_{\min} \in \mathbb{R}_+$ (Pa). Both variables are used operationally.

Operational TC forecasts traditionally rely on physics-based models that solve coupled partial differential equations to simulate global weather fields, with nested higher-resolution grids in the vicinity of the TC (Hazelton et al., 2023). However, we require supercomputing infrastructure to run these models at a spatial resolution that marginally resolves TCs (Davis, 2018) and this limits their accessibility. This limitation, combined with the availability of high-quality, open-sourced benchmark datasets (Rasp et al., 2024), has led to an increased interest in data-driven weather forecasting (Ebert-Uphoff & Hilburn, 2023).

Several "neural weather models" now outperform state-of-the-art physics models on the 1–10 day forecasting of select meteorological fields at a $0.25°$ resolution (Rasp, 2024). Compared to physics-based models, neural weather models—once trained— are computationally efficient and help democratize forecasting. Neural weather models are scalable and ensemble well, which is a desirable property in large forecasts to quantify and reduce uncertainty. Nonetheless, the performance of neural weather models is often summarized in metrics calculated over the global prediction and focus less on

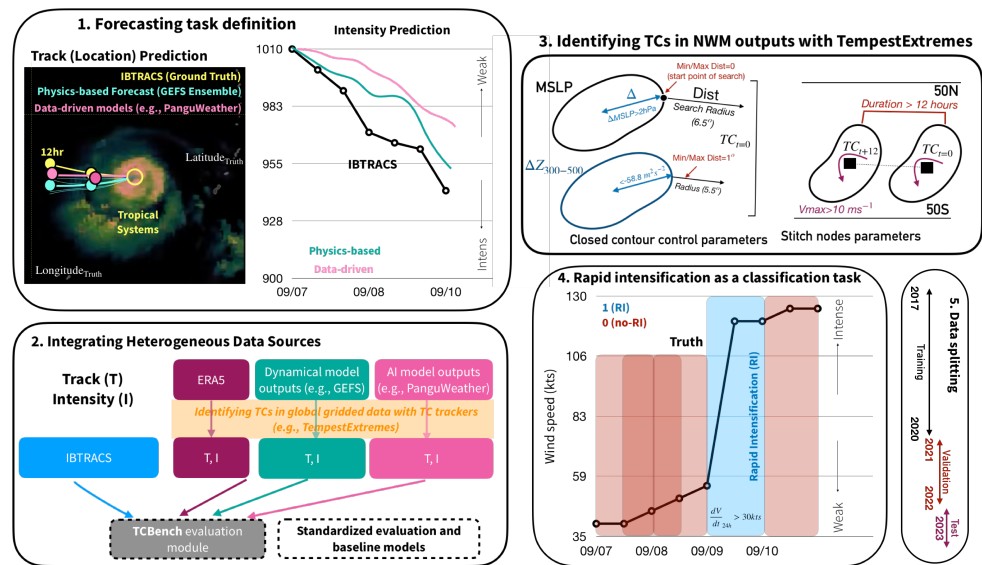

Figure 1: TCBench defines TC forecasting as predicting time-series of track and intensity knowing the system's initial state. It integrates heterogeneous data sources (observations, reanalysis, physics/data-driven models) into a unified evaluation framework to standardize model assessment.

process-based assessments and divergent approaches in evaluation complicates comparison between models. For example, PanguWeather (Bi et al., 2023) and FourCastNet (Pathak et al., 2022) rely on author-defined TC tracking algorithms, GraphCast (Lam et al., 2023) optimizes tracking for each model, and GenCast (Price et al., 2025) relies on a well-established tracking algorithm (Ullrich et al., 2021) but is forced to adapt it due to the frequency of their model outputs. Furthermore, other data-driven approaches that focus on basin specific predictions (e.g., Huang et al., 2023; Park et al., 2023) provide specialized predictions that may outperform global approaches, given the variance in meteorological and oceanographic characteristics that drive TC behavior in each basin(Singh et al., 2025).

While TC track prediction from neural weather models now rivals (DeMaria et al., 2024) and sometimes outperforms physics-based models (Broad, 2024), TC intensity prediction remains a major challenge. We know that intensity change of TCs is primarily determined by the oceanic energy available for the TC to extract and the vertical structure of temperature in the vicinity of the TC (Emanuel, 1986). It is thus possible to derive the theoretical maximum energy achievable by a TC based on sea level pressure, sea surface temperature, relative humidity, and outflow layer temperature (MPI–maximum potential intensity; e.g., Emanuel, 1999). Furthermore, the main processes that keep TCs from intensifying and reaching their theoretical MPIs include vertical wind shear, which introduce dry air intrusion into the TC (Wong & Chan, 2004; Alland et al., 2021), and upwelling of cold deep ocean water underneath the TC (Price, 1981). Though some recent models explicitly predict an ocean variable (e.g., sea surface temperature Price et al., 2025), most predict strictly atmospheric data. Given that the state of the ocean can have a strong impact on TC intensity even under adverse atmospheric conditions (Nickerson et al., 2025), this is likely to present an additional challenge for accurately representing intensity. Furthermore, the spatial resolution of physics-based global models is often too coarse to represent anomalies in wind speed, wind extremes, in TCs (DeMaria et al., 2014; Baker et al., 2024). Similarly, neural models are often forecast low intensity due to biases in their training data (Dulac et al., 2024). As a result, current data-driven intensity forecasts frequently under-perform simple baselines such as persistence and climatology (DeMaria et al., 2024).

Accurate modeling of TC intensity is of special importance, given that Rapid Intensification (RI)—defined as a large increase in intensity over a short period of time, typically around the 95th percentile of intensification in a 24h window—is frequently observed in the most destructive TCs (Lockwood et al., 2024b; Knutson, 2024) and the frequency of such storms is projected to increase with anthropogenic warming Li et al. (2023). Furthermore, RI forecasting remains a major challenge because it arises from nonlinear interactions between environmental forcing and inner-core dynamics Wang

& Wu (2004). These multiscale processes involve inner-core features such as eyewall dynamics, vortex Rossby waves, and rainband structures, along with external influences including vertical wind shear and synoptic flow. The balance between these internal and external factors governs how efficiently a storm can intensify, and their nonlinear coupling across scales makes RI particularly difficult to predict. Yet, the predictive performance of neural weather models for RI has to our knowledge rarely been systematically assessed in previous literature. In this work, we treat RI as a **binary classification task**, which determines whether $V_{max}$ will increase by at least 30 kt ($\approx 15.4$ m s$^{-1}$) over 24 hours (Kaplan & DeMaria, 2003; Kaplan et al., 2010), emphasizing that this roughly corresponds to the 95th percentile of the rate of intensity increase. By formulating RI evaluation in this way, we aim to evaluate how these models could be used as flags in warning systems. This is important given the need for better forecasting of these events to coordinate prompt responses and minimize the impacts associated with these storms (Appendini, 2024).

In this work, we present TCBench, a benchmark dataset that **bridges state-of-the-art neural and physics-based weather models with TC observational records to accelerate progress in data-driven forecasting of TCs**. This enables a transparent, reproducible assessment of how TC prediction can be further improved with machine learning. We demonstrate that neural weather models can skillfully forecast TC intensity up to 5 days ahead, particularly when combined with observational data and processed through tools provided in TCBench. Furthermore, we show that ensemble neural weather models demonstrate TC track prediction skill that rivals traditional, physics-based ensembles.

TCBench frames the problem of TC forecasting under the assumption that there is already a tropical system that exists at an initial time of forecast, thereby focusing on predicting how the system will evolve. TCBench provides pre-processing pipelines, evaluation protocols, visualization tools, and baselines from both physics- and neural-based models to benefit the atmospheric science and AI communities. Designed for rapid iteration and anticipating fast innovation in data-driven forecasting, TCBench is extensible to new models, additional predictive targets, and encourages creative use of the data's spatiotemporal structure.

## 2 RELATED WORK

**Statistical models skillfully forecast TC intensity up to 5 days ahead when combined with physics-based global atmospheric models, a strategy known as "statistical-dynamical" prediction**. A prominent example is the Statistical Hurricane Intensity Prediction Scheme (SHIPS, (DeMaria & Kaplan, 1994; Kaplan & DeMaria, 1999; DeMaria et al., 2005)), which uses a multiple regression model with area-averaged climatological, persistence, satellite observations, and environmental predictors from physics-based models to forecast TC intensity. Statistical-dynamical TC intensity prediction models remain in use operationally due to their skill, particularly at longer forecast lead times (Cangialosi et al., 2020). Statistical-dynamical models are also skillful at predicting RI, unlike purely physics-based models (Torn & DeMaria, 2021). Statistical-dynamical methods for RI forecasts, such as the SHIPS Rapid Intensification Index, or SHIPS-RII (Kaplan et al., 2010; 2015), use linear discriminant analysis (Kaplan et al., 2010; Knaff et al., 2018), Bayesian and logistic regression models (Rozoff & Kossin, 2011; Knaff et al., 2018), binomial logistic regression (DeMaria et al., 2021), as well as consensus models (Kaplan et al., 2015).

In recent years, **machine learning models have begun tackling the problem of TC intensity prediction**. Machine learning models trained on SHIPS predictors and satellite imagery skillfully predict TC intensity and RI (Su et al., 2020; Griffin et al., 2022), as well as intensity and track errors (Barnes et al., 2023; Fernandez et al., 2025). Convolutional neural networks (CNNs) and k-means clustering better distinguish between environments that are favorable and not favorable to RI (Mercer et al., 2021). Other approaches, such as decision tree-based models trained on SHIPS predictors (Shaiba & Hahsler, 2016), the random Forest-based RI Scheme (FRIA) (Slocum, 2021; Sampson et al., 2023), and the Long Short-Term Memory method (Yang et al., 2020) have also shown skill in RI prediction. The advent of neural weather models for the global atmosphere opens new opportunities for data-driven post-processing, analogous to statistical-dynamical prediction, with the **potential to significantly extend the lead time for skillful TC forecasts**.

TCBench targets the challenging problem of global TC forecasting, providing open, real-time–available data for these notoriously difficult-to-predict phenomena in an accessible format designed to accelerate improvement of data-driven models (Table 1). As key contributions from

TCBench to the community, we include routines to deterministically and probabilistically evaluate tracks predicted by models, neural weather model output data associated with our selected test year, and the necessary inputs for the training and validating postprocessing baselines—facilitating evaluation by community members of their own models and approaches with respect to the provided baselines. Furthermore, our forecasting setup provides a straightforward comparison between TC track prediction models to our best observational record, and does not limit comparisons to storms predicted by all models. Finally, we as provide scores for baseline models and thereby streamline comparisons between current and future models.

| Dataset | Forecast evaluation | Global coverage | Open source | TC-specific | Environ. predictors | AI-ready format |
|---|---|---|---|---|---|---|
| WeatherBench 2[1] | ✓ | ✓ | ✓ | ✗ | ✓ | ✓ |
| ChaosBench[2] | ✓ | ✓ | ✓ | ✗ | ✓ | ✓ |
| SHIPS[3] | ✗ | ✗ | ✗ | ✓ | ✓ | ✗ |
| IBTrACS[4] ("ground truth") | ✗ | ✓ | ✓ | ✓ | ✗ | ✗ |
| TC-PRIMED[5] | ✗ | ✓ | ✓ | ✓ | ✓ | ✓ |
| TropiCycloneNet[6] | ✗ | ✓ | ✓ | ✓ | ✓ | ✓ |
| **TCBench** | ✓ | ✓ | ✓ | ✓ | ✓ | ✓ |

Table 1: Comparison of datasets and benchmarks across key criteria: forecast evaluation support, coverage, openness, TC specificity, environmental predictors, and ML readiness.

[1] Rasp et al. (2024); [2] Nathaniel et al. (2024); [3] DeMaria et al. (2022); [4] Knapp et al. (2010); [5] Razin et al. (2023); [6] Huang et al. (2025).

## 3 TCBench Data Toolbox

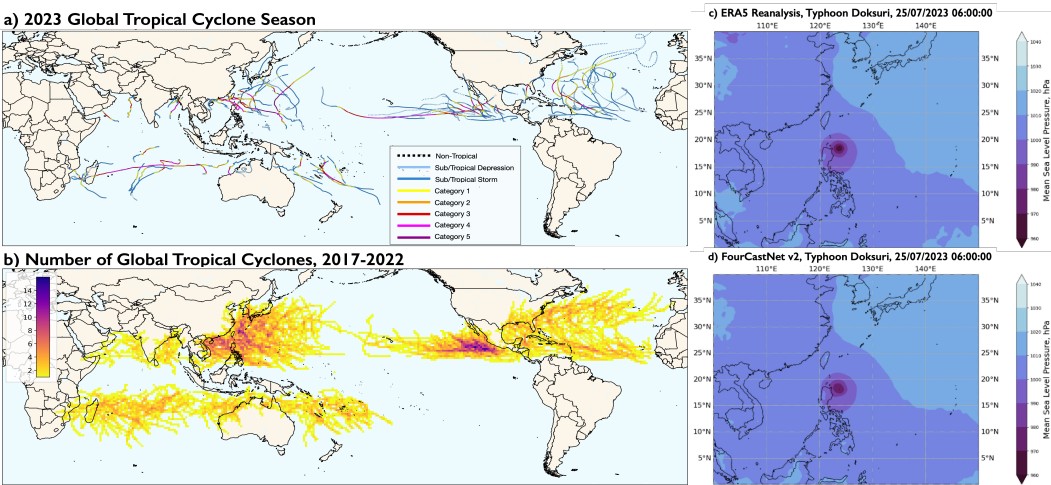

Figure 2: **(a)** 2023 tropical cyclones in the TCBench test year, from IBTrACS. The lines represent the position of each tropical cyclone over time, with the line color representing the storm's intensity at that position. **(b)** IBTrACS estimate of tropical cyclone numbers from 2017-2022 (corresponding to TCBench's training and validation years). Tropical cyclone counts binned into a 1° latitude by 1° longitude grid. **(c)** Mean sea level pressure from ERA5 reanalysis during Typhoon Doksuri, 25/07/2023 06:00:00 UTC. Represents the ground truth targeted by neural weather models, as opposed to the best record of TC intensity (IBTrACS) **(d)** 6-hour forecast of sea level pressure from the FourCastNet-v2 neural weather model during Typhoon Doksuri, valid at 25/07/2023 06:00:00 UTC. Panels a) and b) were made using the troPYcal software package (Burg & Lillo, 2019).

To provide a clear test set and baselines ensuring fair inter-model comparison, TCBench integrates a diverse set of climate data with physics-based and neural weather model forecasts. A summary of

all data sources used in TCBench is provided in Appendix C. These sources were selected based on availability, spatial resolution, historical record length, and their relevance to forecasting or post-processing TC intensity of tracks (Section 3.2). Figure 2 provides an example of some of the various data sources included in TCBench, including observed TC track and intensity data (Figure 2a,b), reanalysis (Figure 2c), and neural weather model forecasts (Figure 2d). In addition to the data we provide, we set forth guidelines for community members to submit data/and or models for their evaluation and listing on the leaderboard.

## 3.1 DATA PROVIDED WITH TCBENCH

**The International Best Track Archive for Climate Stewardship (IBTrACS) serves as the "ground truth" for model evaluation**. IBTrACS is the most complete observational archive of global TCs that is currently available (Knapp et al., 2010; Gahtan et al., 2024). IBTrACS provides global coverage of TC tracks, including various parameters, such as location, intensity, and size. For consistency, we retain only the 6-hourly time stamps (00:00, 06:00, 12:00, 18:00 UTC). We note that the definition of TC intensity is inconsistent across different meteorological agencies (Schreck III et al., 2014). In TCBench, we use the U.S. agencies' definition: minimum sea-level pressure and 1-minute maximum sustained wind speed at an altitude of 10 meters, both of which are provided in IBTrACS. Another source of uncertainty and inconsistency across agencies and time periods is the first data point included in each TC track in the IBTrACS dataset, e.g. there is no specific intensity for when the agencies start the track of a specific TC. We provide tools for processing the IBTrACS file provided by NOAA and provide the subsection of data of interest per identified TC.

**ERA5** (Hersbach et al., 2020) is the 5th generation reanalysis product provided by the European Centre for Medium-Range Weather Forecasts (ECMWF). ERA5 provides dozens of meteorological variables at high spatial, vertical, and temporal resolutions, and models or assimilates a large amount of historical data. Though ERA5 is used to provide initial conditions to physical and neural weather models, we do not provide this data as it is made available by the ECMWF through its Copernicus programme.

**Physics-based models** predict the weather by solving partial differential equations, such as the laws of fluid dynamics, thermodynamics, radiative transfer, and atmospheric chemistry. We include hindcasts from several of these models via The International Grand Global Ensemble (TIGGE) product, including ensemble hindcasts of the Global Ensemble Forecasting System (the GEFS) and the International Forecast System (the IFS)(Bougeault & Coauthors, 2010). In contrast, **neural weather models** make weather forecasts using entirely data driven methods. These methods vary by model but include graph neural networks (Lam et al., 2023), Fourier Neural Operators (Pathak et al., 2022), and vision transformers (Bi et al., 2023). We include hindcasts from several neural weather models, including NVIDIA's FourCastNetv2 (Bonev et al., 2023), Huawei's Pangu-Weather (Bi et al., 2023), Google DeepMind's GenCast (Price et al., 2025), and the ECMWF's single-member AIFS v1.0 (Lang et al., 2024).

With regards to **data processing**, all datasets were reformatted into a unified structure based on IBTrACS for consistency across sources, using the IBTrACS storm identifier for identifying TCs and following unit conventions in IBTrACS. For the physics based models, track data (position, intensity) from international weather prediction centers was extracted from .xml files and standardized to the format. For neural weather model forecasts, forecast tracks and intensities were parsed from model outputs and matched to IBTrACS using TempestExtremes (Ullrich et al., 2021) and HuracanPy (Bourdin & Saffin, 2025) and then standardized to the format. The postprocessing models we provide predict intensification via a parametric distribution, and we thus sample in order to follow the standardized. Additional details for the postprocessing models are provided in Appendix D.4.

## 3.2 DATA INCLUSION CRITERIA

To ensure consistency and usefulness across all experiments, each forecast or model dataset included in TCBench had to meet the following requirements: a) They must provide at least 2 forecast initializations per day (00z, 12z); b) They must provide forecasts at least through a 5-day lead time, or forecasts until the IBTrACS record for that TC ends; c) the forecasts must be available as 6-hourly forecast data; d) the forecasts must provide at least the latitude, longitude, minimum pressure, and maximum wind speed track variables; e) models must provide any accompanying environmental

fields (e.g., surface wind speed, surface pressure, geopotential height; additional examples are given in Appendix C) used for tracking and intensity estimation; f) if tracking storms in environmental fields, a documented tracking algorithm and reproducible code examples; and g) if not directly forecasting track and intensity, the provided fields must have a minimum resolution corresponding to about 0.5° on a global lat-lon grid, preferably 0.25°. These criteria are guidelines for both the data we provide, and data provided by community members in the future.

### 3.3 GUIDELINES FOR MODEL SUBMISSION

**Dataset Split Recommendation:** IBTrACS storms are divided based on the year they occurred to prevent leakage, noting that we divide based on calendar year and not TC season. We suggest at least 4 years be used for training and at least 2 years be used for validation, noting that we assign 2017-2020 to training and 2021-2022 to validation. **We require that the year 2023 be left for testing**, given that the models we include as baselines are not trained or tuned on this year.

**Model Submission Guidelines:** TCBench submissions must include: a) a list of open data sources used for model training, including any relevant links; b) any additional data needed to run the model on the test set; c) the minimal data sources needed to train the baselines presented for the training and validation years, d) code and hyperparameters to train and run the model and generate the predictions, e) the model and/or trained weights in a standard format (e.g., a pickled file – .pkl), f) and an environment file to recreate the computing environment (or a container with which to run the model). All data will be added to the GitHub and HuggingFace repositories as appropriate.

## 4 TCBENCH BENCHMARKING TOOLBOX

TCBench provides a standardized set of metrics to provide objective comparisons for TC forecasting.

**Intensity:** Intensity forecasts are evaluated using the root mean square error (RMSE) and the mean absolute error (MAE).

**Track Error Metrics:** To evaluate the quality of predicted storm tracks, a set of deterministic error metrics is computed following the methodology described by Heming (Heming, 2017). These include:

**Direct Positional Error (DPE):** The straight-line distance between the forecast and observed storm positions at the same verification time.
**Cross-Track Error (CTE):** The component of DPE that lies perpendicular to the observed storm motion, indicating lateral displacement.
**Along-Track Error (ATE):** The component of DPE that lies along the observed storm motion, indicating whether the forecast storm is ahead or behind in time.

A complete description of the formulae and computational procedures is provided in Appendix E.

**Probabilistic Evaluation:** To quantify uncertainty in ensemble forecasts, we report the fair variant of the continuous ranked probability score (CRPS). Given an ensemble of $N$ forecast values $\{x_i\}_{i=1}^{N}$ and an observation $y$, the fair CRPS provides an unbiased estimation of the discrepancy between the forecast distribution and the observed outcome:

$$\text{CRPS}(\{x_i\}, y) = \frac{1}{N} \sum_{i=1}^{N} |x_i - y| - \frac{1}{2N(N-1)} \sum_{i=1}^{N} \sum_{j=1}^{N} |x_i - x_j|. \tag{1}$$

The first term captures ensemble accuracy (mean absolute error), while the second reflects ensemble sharpness (spread). Lower CRPS indicates better-calibrated forecasts centered on the observation. We define **Track CRPS** for TC tracks by replacing absolute differences with Haversine distances between predicted and observed storm center positions.

**Rapid Intensification Evaluation:** Though Rapid intensification (RI) is a rare event, it is not exceedingly so. We define it as approximately the 95th percentile of intensity change over a 24-hour

| Closed Contour Command Parameters | | | | Stitch Nodes Parameters | |
|---|---|---|---|---|---|
| **Variable** | **Delta** | **Dist. (°)** | **Min/Max Dist. (°)** | **Parameter** | **Value** |
| MSLP | 200 Pa | 6.5 | 0 | Minimum Track Duration | 12h |
| $\Delta z_{300,500}$ | -58.8 m$^2$s$^{-2}$ | 5.5 | 1.0 | $V_{\max}$ Threshold | 10 m/s |
| | | | | Latitude ($\phi$) | $|\phi| \leq 50°$ |

Table 2: TempestExtremes Parameters. Distances are expressed in great circle degrees.

window. Thus, we evaluate model performance on RI forecasts using metrics more tailored for rare event forecasts. We note that while most of TCBench is configured as a **regression** problem, we have formulated RI as a **binary classification problem**. Thus, rapid intensification models are tasked with making a simple "yes/no" prediction for the occurrence of rapid intensification (i.e., the presence of an intensification of 30 knots per 24h window, where the window is rolled over the 120h forecasts). For these metrics, TP refers to true positives (hits); TN refers to true negatives, or correct negatives; FP refers to false positives, or false alarms; and FN refers to false negatives, or misses.

**Critical Success Index (CSI)**: Also known as the Threat Score, measures the ratio of correctly predicted positive observations to the sum of all predicted positives, actual positives, and minus true positives. CSI can be used both for probabilistic track evaluation and RI.

$$\text{CSI} = \frac{\text{TP}}{\text{TP} + \text{FN} + \text{FP}},$$

## 5 BASELINES

Following standard evaluation procedures (Knaff et al., 2003), we first compare TC forecasts to the Mean Tendency by Lead & Basin (MT-LB) and persistence baselines. MT-LB samples the empirical distribution of IBTrACS targets (intensity and position changes relative to initial conditions) from 1980 to 2022, conditioned on basin and lead time. For Persistence, at each lead time $L$ the forecast equals the initial state at $t_0$ (i.e., we predict neither motion nor a change in intensity)

**Baselines for Track and Intensity Predictions:** We provide scores for tracks obtained directly from DeepMind's Weather Lab website for GenCast and Weatherlab FNv3, and for the ECMWF IFS ensemble and the NCEP GEFS models using tracks from the TIGGE dataset. For PanguWeather, FourcastNetv2, and the single-member version of AIFS, we process global outputs to obtain tracks using TempestExtremes (Ullrich et al., 2021). The parameters we used are listed in Table 2, and are the same as in Ullrich et al. (2021), with the exception of the minimum track duration, which is set to 12h. While this would normally lead to overly sensitive tracking (i.e., resulting in less filtering of short-lived systems that are not true cyclones), we frame the problem of TC prediction with the assumption that there is a prior system of interest known to exist at an initial time and we are interested in the evolution of said tropical system. We thus eliminate all spurious tracks that could not be matched to the systems of interest. We generate two sets of tracks to evaluate - one in which there is a fall back to the persistence baseline whenever the model does not produce a forecast for a storm (e.g., because the storm does not exist in the model outputs) and one that includes only the storms predicted by the models. Our main results rely on the former, while plots associated with the latter can be found in Appendix F.

We provide additional baselines that post-process data-driven weather model forecasts to predict tropical cyclone intensity, using a pipeline that follows (Gomez et al., 2025). These baselines provide only an intensity forecast, and a more thorough description of these baselines is given in Appendix D.4. Finally, we use all baselines for intensity prediction to evaluate for rapid intensification (RI) by thresholding the wind speed change—i.e., if the 24 hour change in wind speed is at least 30 kts, RI has occurred.

# 6 RESULTS

In this section we present examples of using TCBench to evaluate the deterministic and probabilistic skills of neural weather models (AI Models) and physics-based models on TC track and intensity predictions, covering lead times of up to five days. The results are summarized in Figures 3 and 4.

**Some AI Models perform well for tracks:** As shown in Figure 3, track prediction skills deteriorate with increasing lead time. All evaluated models exhibit track errors that are smaller than the persistence error at all lead times, indicating that all models produce useful track prediction. This is especially apparent when looking at the error for the predicted tracks without a fallback to persistence, shown by Figure 6 in Appendix F. While the track error for most AI models and physics-based models are very similar across lead times, we see a clear indication of the AIFS model to outperform other models in deterministic metrics beyond 24 hours. We largely attribute this to AIFS producing TC track predictions for a greater number of time steps (see Figure 7 in Appendix F ), which greatly reduces the number of samples for which the evaluation falls back to the persistence baseline. Regarding the quality of model track ensembles, the physics-based GEFS model still outperforms the AI model alternatives we evaluated with regards to track displacement CRPS. These results highlight the potential of some existing data-driven forecasting models to provide **reliable deterministic** TC track forecasts, but suggest that they are yet to be as good as physics-based ensemble models in probabilistic track predictions.

**Postprocessing AI models yields skillful intensity predictions:** The coarse spatial resolution of the physics-based global weather prediction models and the underlying training data for the AI models mean that neither type of model is likely to perform well for intensity predictions. In fact, all models produce forecasts of $V_{max}$ and $p_{min}$ that are worse than the persistence baseline for shorter lead times but that outperform it for longer lead times (fig. 3). We see that the lead time at which the intensity predictions become useful varies between 24 hours and 48 hours, and that the physics-based GEFS model tends to perform best for both $V_{max}$ and $p_{min}$. However, the difference in deterministic and probabilistic skills between the best raw AI model and GEFS is slight. Furthermore, we do not discount the possibility that a high spatial resolution, non-global physics-based model can be dramatically better than both GEFS and the AI models at predicting TC intensity. Finally, a well-designed post-processing algorithm of raw AI predictions can improve upon an originally inaccurate AI model (e.g., PanguWeather) to the extent of producing predictions on par with or better than the GEFS model.

**Only the postprocessed AI models capture RI:** The challenges of predicting TC RI events with existing global AI models and physics-based models can clearly be observed in Fig. 4, where most models show little to no ability to forecast such events across different lead times. Of the models we evaluated, only the post-processed PanguWeather model shows any skill at RI prediction, with some success in detecting RI for lead times between 48 and 96 hours. We note that neural weather models have generally been trained on ERA5 reanalysis, which is known to have a negative bias with regards to TC intensity. This issue, combined with the fact that many models have been trained to reduced a mean error (e.g., MAE, RMSE) and the relatively coarse resolution of the predicted fields, makes predictions of rapid intensification by the models a significantly more challenging task.

Overall, our evaluation reveals that:

- Some AI models outperform GEFS in deterministic track prediction, but not probabilistic ones.
- GEFS provides slightly better deterministic and probabilistic intensity forecasts than AI models.
- Intensity forecasts from AI models can be improved with a postprocessing pipeline.

These findings emphasize the complementary strengths of neural and physics-based systems, and point toward hybrid approaches as a promising future direction in TC forecasting. These findings also agree with (DeMaria et al., 2024; Sahu et al., 2025), who found that neural weather models could make skillful TC track predictions, but lacked skill in intensity forecasting. The results from the post-processed neural models are encouraging and point the way towards making data-driven TC intensity forecasts more reliable.

## 7 DISCUSSION AND CONCLUSION

**Other Potential Applications:** In addition to the simple deterministic and probabilistic TC track and intensity predictions presented above, the TCBench dataset can be applied to other scientific applications related to different aspects of tropical cyclones. These include assessments of wind-related risks posed by tropical cyclones, tropical cyclone wind and precipitation nowcasting, wind field reconstruction, and data aggregation for physics discovery. Additional description of these potential applications is provided in Appendix G.

**Limitations:** There are clear limitations regarding this work, including the variance and evolution of uncertainty in the data, a limited selection of sources for tractability, and TC track inconsistencies across agencies. We acknowledge these and expand on them further in Appendix H.

**Conclusion:** By emphasizing flexibility in data experimentation, TCBench provides a data foundation that can be extended with advancements in TC observations and modeling. TCBench's goal is to help the research community improve TC forecasts, our understanding of the physical processes governing TC behavior, ultimately mitigating TC impacts on local communities and society at large. We additionally provide a common point of comparison between TC forecasting models that is centered around the prediction of TC behavior once systems have formed. We show that for this task there is ample space for further research and improvement beyond our current data-driven forecasting capabilities, and provide tools for helping researchers in this field.

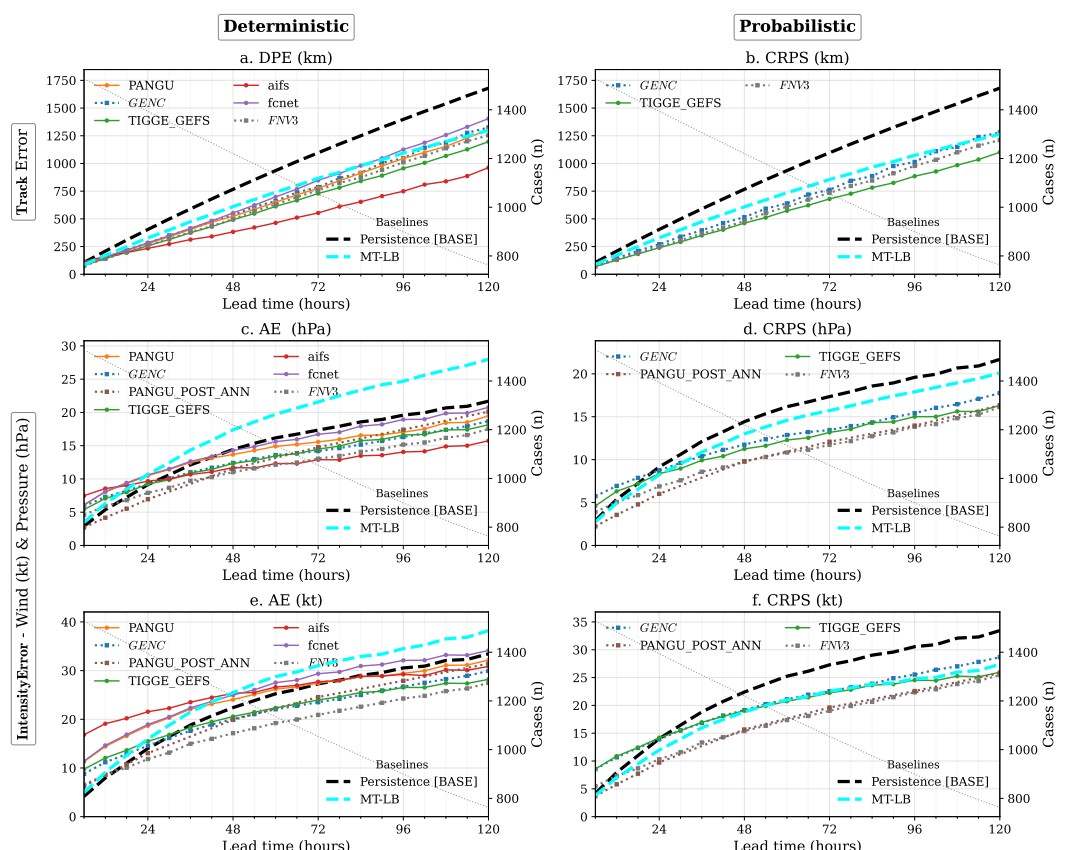

Figure 3: **FAIR per-lead comparison on TCBench-2023.** Deterministic (left) and probabilistic (right) scores from 6–120 h for: **(a)** DPE, **(b)** CRPS–track, **(c)** AE–pressure, **(d)** CRPS–pressure, **(e)** AE–wind, **(f)** CRPS–wind. Means are computed on IBTrACS verification keys (00/12Z inits), with missing entries filled via persistence for fair comparison. Baselines: persistence (black dashed) and **MT-LB** (cyan dashed; mean tendency by lead & basin from IBTrACS, 1980–2022). For CRPS, persistence equals the MAE delta forecast. *GENC* and *FNV3* are italicized because we use the providers' tracks (not re-derived). The post-processing model (dotted; e.g., PANGU_POST_ANN) deviates from our protocol (non-6 h lead grid; trained/validated on different years). Right axes show IBTrACS case counts. See SI for the corresponding raw (non-filled) comparison.

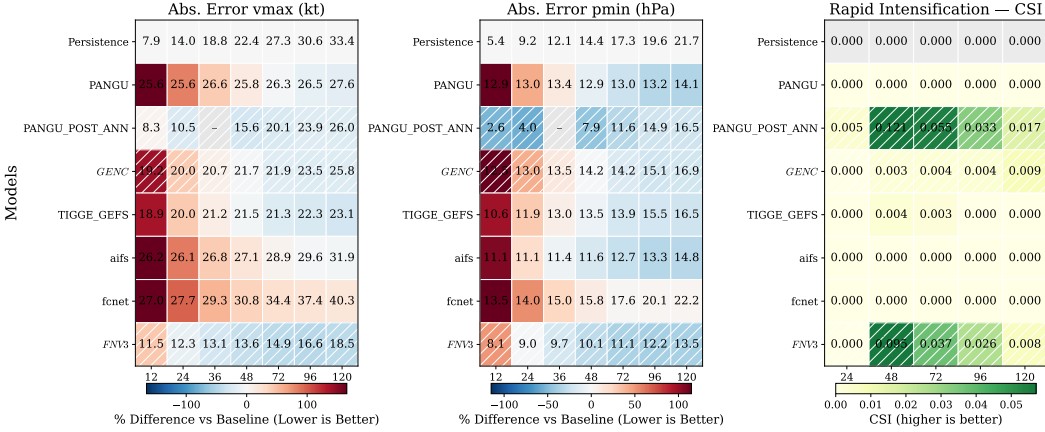

Figure 4: TCBench deterministic scorecard (2023, baseline = Persistence). Cells show mean error; colors show % difference vs baseline at the same lead (12-120h; columns) for each model (rows).

## 8 LLM USAGE DISCLAIMER

Conversational AI tools (e.g., ChatGPT and Copilot) were used for the following tasks during the manuscript's preparation: text editing (specifically, in making certain sections more succinct), latex formatting, and as an aid in code development when writing the benchmark framework.

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

# APPENDICES

## A ACCOUNTABILITY AND REPRODUCIBILITY

TCBench is released under the open-source GNU General Public License. Continued development, including updates discussed in the limitations section, will be managed on the official TCBench GitHub page. Maintenance includes community support, issue tracking, and ongoing curation of benchmark content. All assets are hosted on GitHub and HuggingFace to ensure stability and accessibility.

**Resources:**

- **Dataset:** Available at *redacted for anonymity*
- **Model Checkpoints:** *redacted for anonymity*
- **Code:** *redacted for anonymity*
- **Documentation:** *redacted for anonymity*

An anonymized subset of the code and data has been prepared and uploaded to ProtonDrive:

`https://drive.proton.me/urls/SKT1FGC3KG#lIOFkFcL4xOU`

Please note that though any reference to usernames and paths that might be used to identify the developers were removed, metadata scrubs were not conducted. We request that reviewers refrain from viewing file metadata in case any identifying information be contained within.

### A.1 HUGGINGFACE REPOSITORY STRUCTURE

We organize the TCBench HuggingFace repository as follows:

- texttt2023_IBTrACS.csv: The subset of IBTrACS with the storms and timestamps that we use to evaluate models.
- `matched_tracks/`: A collection of CSVs with the tracks we evaluate as baselines, including AIFS, FourCastNetv2, Weatherlab FNv3, GenCast, Panguweather, TIGGE GEFS, and TIGGE IFS.
- `neural_weather_models/`: Raw, global neural weather model outputs for up to 5 days lead time, initialized at each timestamp associated with a storm in *2023_IBTrACS.csv* (i.e., the ground truth).
- `postprocessing_models/`: Postprocessing model weights and the data required to run them on the test set.

### A.2 GITHUB REPOSITORY STRUCTURE

We organize the TCBench Github repository as follows:

- `track_processing/`: The TempestExtremes routines needed to generate the tracks associated with the neural weather models, and the python routines that rely on HuracanPY to match the tracks found with TempestExtremes to the TCs in IBTrACS.
- `utils/`: Collection of utilities used to process IBTrACS data, to process neural weather model data for post-processing, and to calculate metrics on the tracks.
- `scriptnames.py`: various python scripts used to generate the track baseline predictions (e.g., *baselines.py, climatology_maker.py, compute_persistence.py*), to evaluate a set of tracks (e.g., *evaluate_tracks.py*)

## B GETTING STARTED

We provide a detailed description of how to prepare the necessary data, perform training, and benchmark your own model on our website. Please visit the following URL to see the guide: *redacted for anonymity*. We also ask that if you encounter any issues to please feel free to contact us or raise an issue on GitHub.

A getting started notebook for the anonymized review was prepared and is available at:
`https://drive.proton.me/urls/SKT1FGC3KG#lIOFkFcL4xOU`
under the filename *Getting_Started.ipynb*

## C TCBENCH DATA ORGANIZATION AND WORKFLOW

### C.1 DATA SOURCES

TCBench integrates multiple observational, reanalysis, forecast model, and AI-based data sources relevant to tropical cyclone (TC) analysis and prediction. Table S1 summarizes datasets currently included, and points to additional datasets of relevance.

**Atmospheric Field Data:** We use atmospheric field data from a variety of sources, including:

- Climate reanalysis and forecast datasets including ERA-5, TIGGE, IFS, and GFS.
- Outputs from AI models, including PanguWeather, FourcastNet, and the single member version of AIFS 1.0.

### C.2 DATA WORKFLOW

The TCBench workflow standardizes diverse data sources into a common format for evaluation:

1. Forecast and reanalysis fields are subset in space (storm-centered region) and time (forecast lead times) using storm initialization from IBTrACS.

2. Model-specific track outputs (e.g., TIGGE XML files, neural weather model forecasts in netCDF format) are converted into a uniform CSV format that includes the storm identifier (taken from IBTrACS), position, maximum sustained wind, and minimum sea-level pressure.

3. A `TCTrack` object is created for each storm, encapsulating observed and predicted values across time steps.

4. Gridded predictors (e.g., neural weather model forecasts) are stored as multidimensional arrays `[samples, time, lat, lon, variables]` for use in ML-based experiments.

### C.3 DATA PROCESSING

Each dataset undergoes preprocessing to ensure comparability:

| Data Source | Description | Website | Provided |
|---|---|---|---|
| **Reanalysis** | | | |
| ERA5 | European Re-Analysis 5 | https://cds.climate.copernicus.eu/datasets/reanalysis-era5-pressure-levels?tab=overview | No |
| **Operational Analysis** | | | |
| HRES Analysis | ECMWF's archive of operational analysis, distributed through MARS | https://confluence.ecmwf.int/display/UDOC/MARS+content#MARScontent-Atmosphericmodels#Analysis:~:text=models-,Analysis | No |
| **Ground Truth** | | | |
| IBTrACS | Best Track Archive | https://www.ncdc.noaa.gov/ibtracs/ | Yes |
| Extended Best-Tracks | TC tracks with size | https://rammb2.cira.colostate.edu/research/tropical-cyclones/tc_extended_best_track_dataset/ | No |
| TC PRIMED | ML Ready Benchmark dataset | https://rammb-data.cira.colostate.edu/tcprimed/ | No |
| **Statistical Models** | | | |
| SHIPS | Statistical Hurricane Intensity Prediction Scheme (developmental dataset) | https://rammb2.cira.colostate.edu/research/tropical-cyclones/ships/development_data/ | No |
| **Models** | | | |
| NCEP GFS | Global Forecast System | https://www.nco.ncep.noaa.gov/pmb/products/gfs/ | No |
| TIGGE GFS | NCEP GEFS ensemble model tracks dataset | https://rda.ucar.edu/datasets/d330003/dataaccess/# | Yes[†] |
| TIGGE IFS | ECMWF IFS ensemble model tracks dataset | https://rda.ucar.edu/datasets/d330003/dataaccess/# | Yes[†] |
| PanguWeather | AI global model outputs | https://github.com/198808xc/Pangu-Weather | Yes |
| FourCastNet | AI model outputs | https://github.com/NVlabs/FourCastNet | No |
| FourCastNetv2 | AI model outputs | https://github.com/NVIDIA/torch-harmonics | Yes |
| AIFS | AI global forecasting | https://arxiv.org/abs/2406.01465 | Yes |

Table S1: Data Sources Summary. † non-gridded data only

- **TIGGE ensembles:** Extract ensemble mean and member tracks, harmonize to IBTrACS temporal resolution.
- **AI models:** Post-process global forecasts to storm-centered tracks; derive intensity and MSLP comparable to IBTrACS.
- **Post-processing Models:** Extract 50 member ensemble from parametric distribution through sampling, clip predictions to physical values ($V_{max} \geq 0\ kt$).

| Environmental Variable | Pressure Level (hPa) | | | | | | | | | | | | | | | | |
|---|---|---|---|---|---|---|---|---|---|---|---|---|---|---|---|---|---|
| | 1000 | 975 | 950 | 925 | 900 | 850 | 800 | 700 | 600 | 500 | 400 | 300 | 250 | 200 | 150 | 100 | 50* |
| Relative Vorticity | ✓ | ✓ | ✓ | ✓ | ✓ | ✓ | ✓ | ✓ | ✓ | ✓ | ✓ | ✓ | ✓ | ✓ | ✓ | ✓ | ✓ |
| Relative Humidity | ✓ | ✓ | ✓ | ✓ | ✓ | ✓ | ✓ | ✓ | ✓ | ✓ | ✓ | ✓ | ✓ | ✓ | ✓ | ✓ | ✓ |
| Geopotential Height | ✓ | ✓ | ✓ | ✓ | ✓ | ✓ | ✓ | ✓ | ✓ | ✓ | ✓ | ✓ | ✓ | ✓ | ✓ | ✓ | ✓ |
| Vertical Velocity | ✓ | ✓ | | ✓ | | ✓ | | ✓ | ✓ | ✓ | ✓ | ✓ | ✓ | ✓ | ✓ | ✓ | |
| Horizontal Divergence | ✓ | | | ✓ | | ✓ | ✓ | ✓ | ✓ | ✓ | ✓ | ✓ | ✓ | ✓ | | ✓ | ✓ |
| Equivalent Potential Temperature | ✓ | | | ✓ | | ✓ | | ✓ | ✓ | ✓ | ✓ | ✓ | ✓ | ✓ | | | |
| Zonal Wind (u) | ✓ | | | ✓ | | ✓ | | ✓ | | ✓ | | ✓ | ✓ | ✓ | | | |
| Meridional Wind (v) | ✓ | | | ✓ | | ✓ | | ✓ | | ✓ | | ✓ | ✓ | ✓ | | | |

Table S2: Pressure level environmental fields potentially useful for TC intensity prediction. Adapted from (Ganesh Sudheesh et al., 2023).

| | |
|---|---|
| Temperature (2m) | Dew point temperature (2m) |
| Convective available potential energy | Sea surface temperature |
| Total column water vapor | Total column cloud ice water |
| Total column cloud liquid water | Total column super-cooled liquid water |
| Total column cloud rain water | Vertical integral of divergence of cloud frozen water flux |
| Vertical integral of divergence of cloud liquid water flux | Vertical integral of divergence of mass flux |
| Vertical integral of divergence of moisture flux | Vertical integral of divergence of total energy flux |
| Vertical integral of potential and internal energy | |

Table S3: Single level environmental fields potentially useful for TC intensity prediction. Adapted from (Ganesh Sudheesh et al., 2023).

## C.4 Model Evaluation and Metrics

We evaluate forecast performance using the following metrics, standard in tropical cyclone verification:

- **Track error:** Great-circle distance (km) between forecast and observed positions.
- **Intensity error:** Mean absolute error (MAE) of maximum sustained winds (kt) and minimum sea-level pressure (hPa).
- **Skill scores:** Relative to climatology, persistence, and SHIPS baselines.
- **Probabilistic metrics:** Brier score and reliability diagrams for ensemble/AI probability forecasts.
- **Partitioning:** Cross-validation by year ensures temporal independence between training and evaluation sets.

## C.5 Biases and Limitations

- The dataset represents a subset of all possible TC tracks; some regions and seasons may be underrepresented, especially where fewer storms materialize.

- Forecast evaluation may omit rare teleconnection patterns or unusual environmental scenarios, limiting generalizability.
- Definitions of storm intensity (e.g., 1-min vs. 10-min sustained winds) vary by agency and observation method, introducing potential inconsistencies across basins. We control for this by relying on USA reported values.
- IBTrACS quality varies by basin, with lower reliability in the South Indian Ocean and in the pre-satellite era.
- Reanalysis products such as ERA5 contain uncertainties in poorly observed regions and for weak or short-lived systems.
- AI models are rapidly updated (e.g., FourCastNet v1 vs. v2 vs. v3); benchmark results reflect the versions available to the authors at the time of dataset creation.
- Storm size and structural parameters are inconsistently available across sources, limiting evaluation beyond track and intensity.

# D  BASELINE MODELS

TCBench provides a set of baseline models including traditional statistical approaches, numerical weather prediction (NWP) models, and simple machine learning (ML) approaches. These baselines serve as reference points for track, intensity, and rapid intensification forecasts.

## D.1  PERSISTENCE AND CLIMATOLOGY BASELINES

We rely on persistence and climatology as so called "naïve" baselines to provide context for the performance of the evaluated models against straightforward, transparent prediction methods.

- **Persistence:** Assumes that storm position, intensity, and structure remain unchanged from the current state. Provides a naive forecast useful for short-lead comparisons.
- **Climatology / Mean Track – Long Baseline (MT-LB):** Uses historical averages of storm positions and intensities within each basin and season to generate forecasts. Captures long-term trends and typical seasonal behavior.

## D.2  NUMERICAL WEATHER PREDICTION (NWP) BASELINES

We rely on physics-based Numerical Weather Prediction models as baselines for global predictions of TCs. These models rely on physics and numerical solutions of known equations to predict the state of the atmosphere based on initial conditions. We use the following models in TCBench:

- **NCEP GFS / GEFS:** Global Forecast System provides deterministic forecasts, while GEFS provides ensemble reforecasts from TIGGE. Baselines include position, maximum wind, and minimum sea-level pressure.
- **ECMWF IFS / TIGGE IFS:** Ensemble forecasts from ECMWF's Integrated Forecasting System. Used for both track and intensity benchmarks.

We take the data provided by TIGGE as it is openly available and due to computational limitations associated with running ensembles of physics-based NWP models.

## D.3  NEURAL WEATHER MODEL BASELINES

We rely on neural weather models available at the time of writing to provide baselines for global weather prediction using artificial intelligence. A brief description of the models is given below:

- **FourCastNetV2** - neural weather model that relies on spherical harmonics based neural operators to learn a grid-invariant evolution of the state of the atmosphere.
- **PanguWeather** - neural weather model that relies on an earth-specific transformer architecture and multiple, lead time specific models to predict the evolution of the atmosphere.
- **AIFS Single 1.0** - neural weather model whose architecture relies on a graph-based latent representation of the earth and a sliding window transformer processor in order to predict the

evolution of the atmosphere.

- **Weathernext Gen (GenCast)**[†] - neural weather model that relies on a conditional diffusion architecture to predict a possible evolution of the state of the atmosphere.
- **FNv3**[†] - neural weather model that is not currently open, but is purported to be a functional generative network (FGN) (Alet et al., 2025) fine-tuned on IBTrACS to better predict tropical cyclones.

Track and intensity forecasts for the models marked with †are taken from Google WeatherLab (`https://deepmind.google.com/science/weatherlab`), due to either computational limitations for running the models or to the models not being open at the time of writing.

## D.4 POST PROCESSING BASELINES

We provide a set of postprocessing baseline models following Gomez et al. (2025). Each postprocessing model uses square, clipped neural weather model fields centered on the position of the tropical cyclone at the initial time of forecast, with an extent of +/- 30°latitude and longitude. These models further use an embedding of the position, maximum wind speed, and minimum sea level pressure at the initial time of forecast for their prediction. These models are set up for distributional regression of the rate of the intensification, and output the mean and standard deviation for a gaussian distribution associated with forecast. Of particular note, the postprocessing models only predict intensification, and do not predict the motion of the storm (i.e., the evolution of the position of the TC).

We provide three postprocessing models with varying levels of computational complexity:

- **Multiple Linear Regression:** the neural weather model forecasts are reduced (e.g., by calculating extrema) and a pair of MLRs are trained to predict the mean and standard deviation of the rate of intensification.
- **Multilayer Perceptron (ANN):** the neural weather model forecasts are reduced in a manner consistent with the reduction used for the MLR, but the regression algorithm is instead a MLP.
- **UNet:** the neural weather model forecasts and an embedding of the scalar quantities associated with the forecast (e.g., the position and intensity at the initial time of forecast) are used in the regression task.

We use the details provided for these architectures and associated hyperparameters provided by Gomez et al. (2025) in training the models.

## D.5 TRAINING SETUP

All baselines are trained and evaluated consistently:

- **Data Splits:** Storms are split into folds by year to minimize temporal leakage, ensuring that models are evaluated on unseen storms.

- **Input Variables:** Depending on the baseline, inputs include:

  - Current storm state: latitude, longitude, maximum wind, minimum pressure

  - Environmental predictors: derived from ERA5 or TIGGE

  - Historical storm statistics (for climatology/MT-LB)

- **Ensemble Strategy:** For probabilistic evaluation, ML ensembles produce multiple realizations; NWP ensembles (e.g., GEFS, IFS) are used directly as provided; Post-processing model output distributions are sampled to produce a 50-member ensemble.

Baseline forecasts are evaluated using the same deterministic, probabilistic, and rare event metrics described in Section 4.

## D.6 Baseline Models Summary Table

| Baseline | Type | Inputs | Outputs / Target |
|---|---|---|---|
| Persistence | Deterministic | Current storm state | Track, intensity |
| Climatology / MT-LB | Statistical | Historical storm averages | Track, intensity |
| NCEP GFS / GEFS | NWP / Ensemble | Model initial conditions | Track, intensity |
| TIGGE IFS | NWP / Ensemble | Model initial conditions | Track, intensity |
| FourCastNetv2 | AI | Model Initial Conditions | Track, intensity |
| PanguWeather | AI | Model Initial Conditions | Track, intensity |
| AIFS Single | AI | Model Initial Conditions | Track, intensity |
| WeatherNext Gen | AI | N/A[*] | Track, intensity |
| FNv3 | AI | N/A[*] | Track, intensity |
| Pangu+MLR | PostProcessing AI | AI Model Forecasts, Current Storm state | Intensity |
| Pangu+ANN | PostProcessing AI | AI Model Forecasts, Current Storm state | Intensity |
| Pangu+UNet | PostProcessing AI | AI Model Forecasts, Current Storm state | Intensity |

Table 4: Summary of baseline models included in TCBench. Each baseline serves as a reference for model evaluation.*Forecasts taken directly from Google WeatherLab

# E Evaluation Metrics

To test the forecast performance, TCBench use both deterministic and probabilistic metrics. Deterministic evaluation focuses on the accuracy of predicted intensity and track positions. Standard regression metrics include the root mean square error (RMSE), mean absolute error (MAE), and the coefficient of determination ($R^2$). Track errors are quantified using direct positional error (DPE), cross-track error (CTE), and along-track error (ATE), which decompose the forecast position error along and across the observed storm motion.

For probabilistic forecasts, we evaluate ensemble predictions using the Continuous Ranked Probability Score (CRPS), which measures the discrepancy between the forecast distribution and the observed outcome, capturing both ensemble accuracy and spread. For track forecasts, CRPS is generalized by replacing absolute differences with Haversine distances between forecast and observed storm centers. Additional probabilistic metrics include Brier Skill Score (BSS), Continuous ranked probability score (CRPS), that provide complementary assessments of calibration and uncertainty representation. All details of metrics and computational procedures for all metrics are provided in this section.

## E.1 Deterministic Metrics

- Root Mean Square Error (RMSE): Measures the square root of the average squared differences between predicted and actual values, indicating the model's prediction accuracy.

- Mean Absolute Error (MAE): Represents the average absolute differences between predicted and actual values, reflecting the magnitude of errors in predictions.

- R-squared Score ($R^2$): Quantifies the proportion of the variance in the dependent variable that is predictable from the independent variables, indicating the model's explanatory power.

**Direct Positional Error (DPE)** is the most basic measure of the positional accuracy of a tropical cyclone (TC) forecast. It is defined as the distance between the forecast and observed positions of the storm at the same *verification time (VT)*.

There are two commonly used approaches for calculating DPE:

- **Great Circle Distance (GCD)-based DPE:** This version computes the shortest distance over the Earth's surface between the forecast and observed positions, treating Earth as a

sphere. The *Haversine formula* is commonly used:

$$d = 2r \arcsin\left(\sqrt{\sin^2\left(\frac{\Delta\phi}{2}\right) + \cos(\phi_1)\cos(\phi_2)\sin^2\left(\frac{\Delta\lambda}{2}\right)}\right)$$

where:

- $\phi_1$, $\phi_2$ are the latitudes of the observed and forecast points (in radians),
- $\Delta\phi = \phi_2 - \phi_1$, $\Delta\lambda = \lambda_2 - \lambda_1$ are the latitude and longitude differences,
- $r$ is the Earth's radius (typically $\approx 6371$ km).

This method accounts for Earth's curvature and is the standard in operational TC verification.

- **Cartesian Projection-based DPE:** An alternative approach involves projecting both observed and forecast positions onto a local tangent plane (e.g., using an azimuthal equidistant or equirectangular projection). The DPE is then computed using standard Euclidean distance:

$$d = \sqrt{(x_{\text{fcst}} - x_{\text{obs}})^2 + (y_{\text{fcst}} - y_{\text{obs}})^2}$$

where $(x, y)$ are the projected coordinates. This method may be preferred for regional studies or error decomposition in zonal/meridional directions.

**Note:** DPE provides a scalar magnitude of error but does not convey directionality (e.g., north/south or ahead/behind). Directional errors are captured by the metrics below.

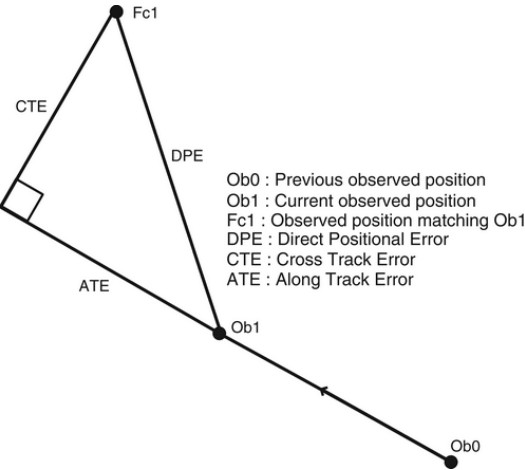

Figure 5: Illustration of track error metrics adapted from (Heming, 2017)

**Cross-Track Error (CTE)** measures the component of the track forecast error *perpendicular* to the observed direction of storm motion.

1. Define the observed motion vector from the storm position 12 hours prior to VT to the observed position at VT.
2. Draw a perpendicular line from the forecast position to this vector.
3. Find the intersection point of this perpendicular with the observed vector.
4. CTE is the great circle distance between the forecast position and this intersection point.

- In the **Northern Hemisphere**, a positive CTE indicates the forecast is to the **right** of the observed path.
- In the **Southern Hemisphere**, the interpretation is reversed.

*Note:* CTE is undefined at the first valid time (due to lack of a 12-hour prior observed position).

**Along-Track Error (ATE)** ATE measures the component of the error *along* the direction of storm motion and reflects timing accuracy.

- Using the same intersection point from the CTE calculation, compute the great circle distance between the observed position at VT and the intersection point.

- Positive ATE indicates the forecast is **ahead** of the observed position (i.e., storm is moving too fast).

- Negative ATE indicates the forecast is **behind** (i.e., storm is too slow).

### E.2    PROBABILISTIC METRICS

Here, we consider an ensemble of $N$ forecasts as a set denoted $\{x_i | i = 1, \ldots, N\}$, and $\{x_i\}_N$ for short.

The **Brier skill score (BSS)** (Brier, 1950) is a relative measure that evaluates the improvement of a probabilistic forecast over a reference forecast or climatology. BSS ranges $-\infty$ to 1, where 1 indicates a perfect skill and 0 indicates no improvement over the reference. A negative BSS suggests that the forecast is less accurate than the reference. The BSS is based on the Brier Score (BS), a metric used to measure the accuracy of probabilistic predictions for binary outcomes, defined as

$$\text{BS} = \frac{1}{N} \sum_{i=1}^{N} (x_i - o_i)$$

where $x_i$ is a forecast probability, $o_i$ is the actual outcome, and $N$ is the number of forecasts. Then BSS is defined as

$$\text{BSS} = 1 - \frac{\text{BS}_{\text{forecast}}}{\text{BS}_{\text{reference}}}$$

The **Continuous Ranked Probability Score (CRPS)** is widely used to assess the skill of ensemble forecasts by measuring the discrepancy between the predicted and observed cumulative distribution functions (CDFs). Unlike the BS, which applies to binary events, the CRPS evaluates the accuracy of probabilistic forecasts for continuous variables. Formally, for a forecast CDF $\widehat{F}$ and a realized value $y$, the CRPS is

$$\text{CRPS} = \int_{\mathbb{R}} \left[ \widehat{F}(x) - \mathcal{H}(x - y) \right]^2 dx, \tag{2}$$

where $\mathcal{H}$ denotes the Heaviside step function, which represents the degenerate CDF of a single observation. For discrete ensembles, we use the kernel representation of the CRPS (Gneiting & and, 2007):

$$\text{CRPS}(\{x_i\}_N, y) = \frac{1}{N} \sum_{i=1}^{N} |x_i - y| \; - \; \frac{1}{2N^2} \sum_{i=1}^{N} \sum_{j=1}^{N} |x_i - x_j|. \tag{3}$$

The first term is the mean absolute error of the ensemble members, while the second term measures the ensemble spread via the mean absolute pairwise difference. To mitigate finite-ensemble bias (inflated scores due to underestimation of the second term), we adopt the fair CRPS (Zamo & Naveau, 2018):

$$\text{fCRPS}(\{x_i\}_N, y) = \frac{1}{N} \sum_{i=1}^{N} |x_i - y| \; - \; \frac{1}{2N(N-1)} \sum_{i=1}^{N} \sum_{j=1}^{N} |x_i - x_j|. \tag{4}$$

### E.3    METRICS FOR EXTREMES

**Rapid Intensification:** Rapid intensification is a rare event, but not an exceedingly rare event; by definition, it is the 95th percentile of intensity change. Thus, we evaluate model performance on rapid intensification forecasts using metrics more tailored for rare event forecasts. We note that while most of TCBench is configured as a **regression** problem, we have formulated rapid intensification as a **binary classification problem**. This means rapid intensification models will be tasked with making a simple "yes/no" prediction for the occurrence of rapid intensification.

- Critical Success Index (CSI): Also known as the Threat Score, measures the ratio of correctly predicted positive observations to the sum of all predicted positives, actual positives, and minus true positives. Can be used both for probabilistic track evaluation and RI.

- Peirce skill score (PSS): The Peirce skill score (also known as the Hansen and Kuipers discriminant, or the true skill statistic) is an estimate of how well the forecast separates "yes" events from "no" events. It ranges from -1 to 1, with +1 being a perfect score and 0 indicating no skill. PSS is generally considered to be better for rare events, though for extremely rare events it tends to 0.

$$\text{PSS} \;=\; \text{TPR} - \text{FPR} \quad \text{where TPR} \;=\; \frac{\text{TP}}{\text{TP}+\text{FN}}, \quad \text{FPR} \;=\; \frac{\text{FP}}{\text{FP}+\text{TN}}. \quad (5)$$

# F    EXTENDED RESULTS

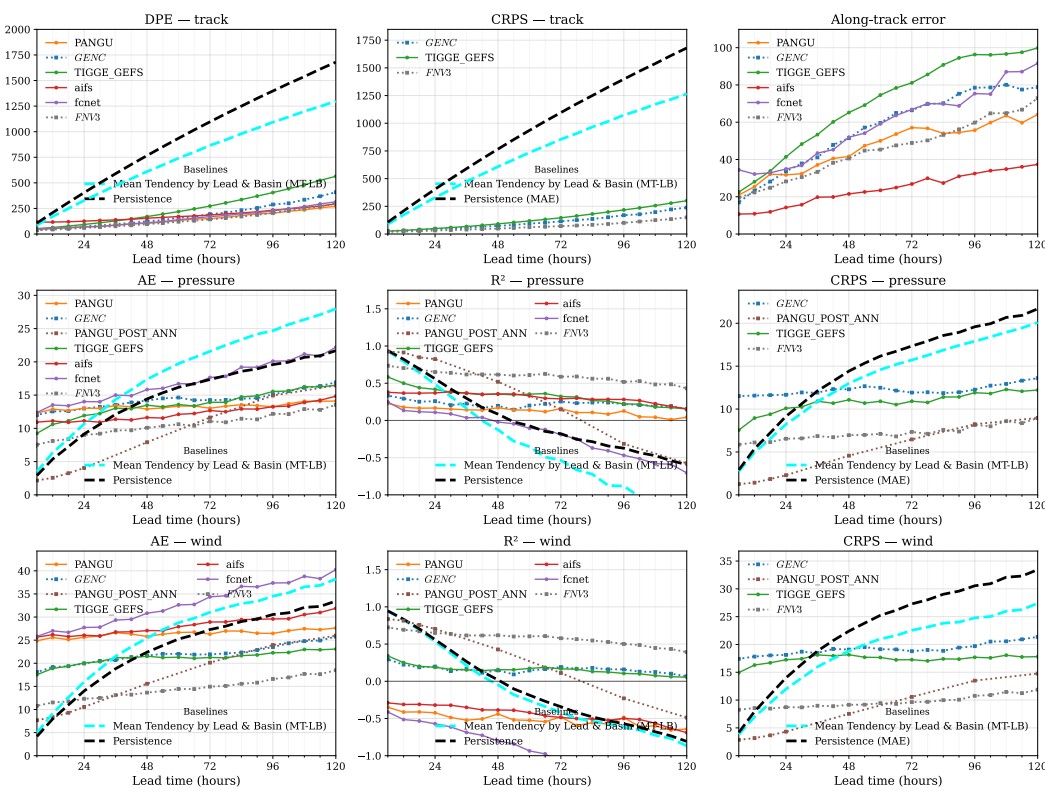

Figure 6: **Per–lead verification on TCBench-2023 (6–120 h), non-filled.** Rows: track, pressure, wind; columns: AE, $R^2$, CRPS (track uses DPE, CRPS-track, along-track). Curves use *raw model coverage*—no persistence filling—so each mean is over the forecast–verification pairs available for that model and lead. Models are colored (WeatherLab FNv3 in gray). Baselines: persistence (black dashed) and climatology/MT-LB (cyan dashed). $R^2 = 1 - \text{MSE}/\text{Var}$ vs. IBTrACS

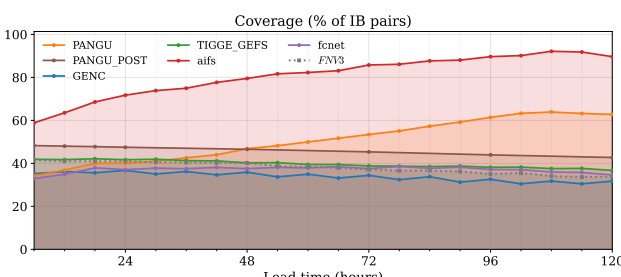

Figure 7: **Coverage on the 2023 test set (% of IBTrACS verification pairs) by lead.** An IBTrACS pair is a unique $(SID, t_0, t_0+L)$ observed on the 6-hour grid with $t_0 \in \{00, 12\}$ UTC. A model *covers* a pair if it outputs any row for that key. Shaded regions indicate the fraction covered at each lead. Large coverage disparities motivate the FAIR comparison (persistence-filled on the same IB grid) reported elsewhere in the paper.

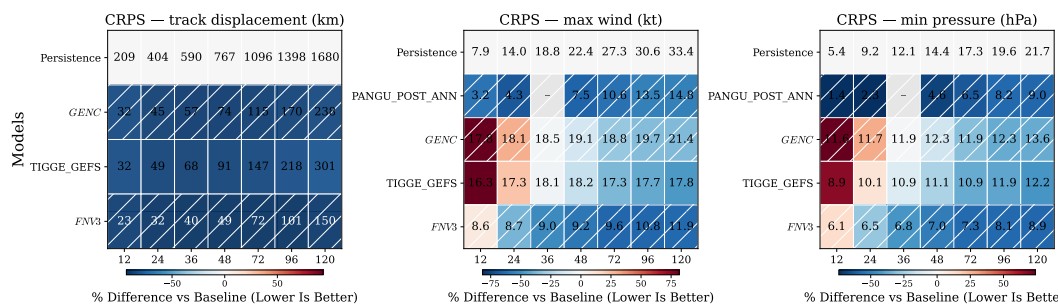

Figure 8: **CRPS scorecards (probabilistic).** Track displacement (km), max wind (kt), and min pressure (hPa) on TCBench-2023 for 6–120,h leads. Each cell is the percent difference in CRPS relative to the Persistence baseline at the same lead (lower is better); the Persistence row reports absolute CRPS (units in panel titles). Results use *non-filled* data (raw model coverage); dashes denote no verification pairs at that lead. Hatched rows mark externally provided products (*GENC*, *FNv3*); PANGU_POST_ANN is a learned post-processor.

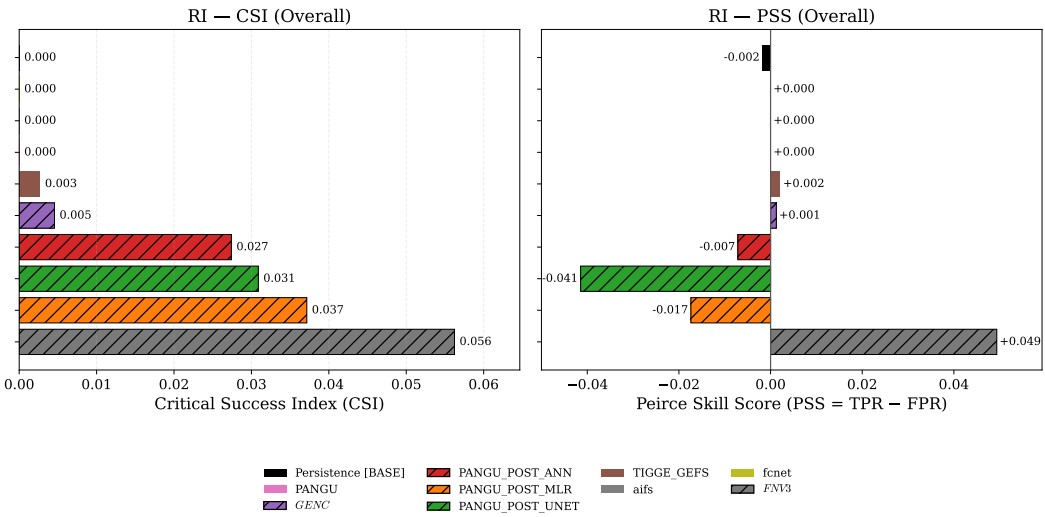

Figure 9: **Rapid Intensification (RI) skill—overall by model.** Bars show overall Critical Success Index (CSI; left) and Peirce Skill Score (PSS = TPR−FPR; right) computed against the IBTrACS RI ground truth for 2023. Scores use the common $(SID, t_0, t_0+L)$ key set on the 6 h IBTrACS grid.

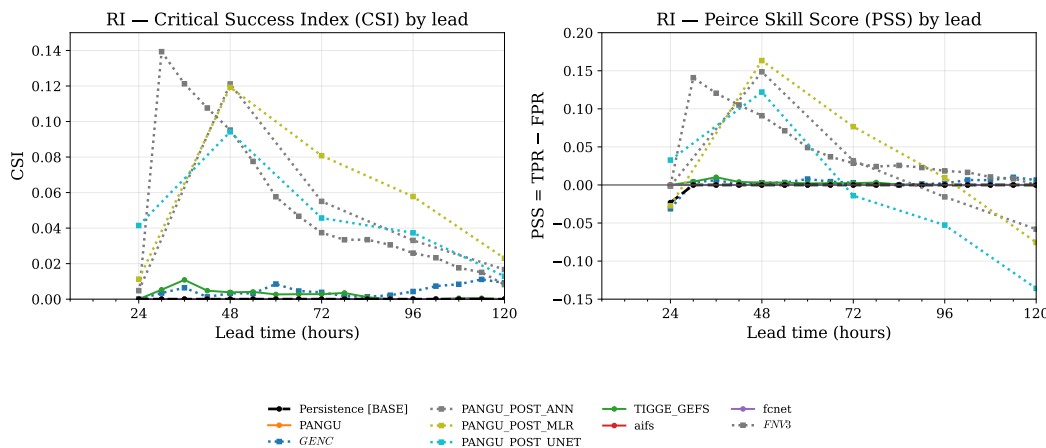

Figure 10: **Rapid Intensification (RI) skill by lead time.** CSI (left) and PSS (right) as a function of lead time (6–120 h, 6 h steps), evaluated vs. the IBTrACS 2023 RI ground truth on the 6 h grid. Axes are capped for readability.

# G  POTENTIAL APPLICATIONS

Accurate assessments of the wind-related risks posed by tropical cyclones rely heavily on knowledge of their full wind structures (2D images stacked in the vertical). It is challenging to obtain such wind fields because satellite observations often only provide partial coverage of TCs with limited temporal sampling. Various methods have been proposed to address this challenge, ranging from analytical parametric models (Chavas et al., 2015) to modern machine learning techniques (Yang et al., 2022), which attempt to infer spatial wind fields from easily observed scalar quantities such as minimum sea-level pressure or the radius of maximum surface winds. By offering IBTrACS in an AI-ready format, TCBench facilitates the design and training of models to **predict key scalars needed for wind reconstruction**. Moreover, the standardized aggregation of outputs from multiple AI models allows users to assess and compare tropical cyclone size predictions across models, thereby helping to identify and quantify TC size biases in neural weather models.

By providing different routines to process TC observations in different formats, the TCBench dataset is ideal for **developing very short-term TC winds and precipitation predictions**. Data-driven nowcasting models (Agrawal et al., 2019) are more computationally competitive than explicitly simulating the TCs with high-resolution numerical models. The ability of probabilistic data-driven models to create hundreds of possible predictions for a given initial condition within minutes and adjust based on new satellite image inputs makes them useful additions to the existing weather forecasting pipeline.

The examples provided in the previous sections only evaluate the intensity at a given point in the TCs. Similar to the wind field reconstruction tasks described earlier, we can **develop wind downscaling models** (Lockwood et al., 2024a) to add additional spatial details to the TCs in the neural weather model outputs in coarse resolution. Another extension to the prediction tasks shown in the text is to provide **global prediction of TC activity across timescales**. The neural weather model outputs and simple tracking algorithm in TCBench facilitate consistent evaluation of the skill of different AI models in developing the TCs at the right time and right place. Based on the comparison of TC activity patterns of different neural weather models, basin-specific refinements can be made to reduce potential spatial biases in the existing trained neural weather models.

Finally, the aggregation of satellite images, environmental fields, and intensity observations in an easy-to-access dataset enables **identification of critical predictive environmental properties for TC intensity (Bister & Emanuel, 2002)**, thereby forming the basis for novel discovery on tropical cyclone physics.

# H  LIMITATIONS

First, uncertainties associated with each product vary, which pose a challenge in estimating and providing accurate error margins for reanalyses and observations. An ambitious goal is to include observational error estimates (e.g., intensity, track location) that vary over time, reflecting improvements in satellites and technology.

Although using multiple reanalyses and tracking algorithms is ideal due to the variation in data quality and sensitivity of cyclone detection schemes, initial efforts may need to rely on a single reanalysis dataset for tractability, while clearly acknowledging its limitations (Pinheiro et al., 2020). Furthermore, creating a single dataset that fits all purposes is impossible, necessitating trade-offs such as neglecting some teleconnections or using datasets only available for the past approximately 15 years. Additionally, the definition of TC intensity varies by agency, primarily in the number of minutes used to calculate maximum sustained wind speeds, which introduces inconsistency.

The first position of TCs across agencies and across time is also subject to large uncertainty and inconsistency. Therefore, in subseasonal to seasonal time-scales and climate studies, it is common to consider the first time the TC reaches tropical storm intensity (35 kt) in the track to improve consistency. Moreover, creating a seamless list of predictors for the entire tropics is challenging due to the variation in relevant variables across different basins or seasons.

