# OpenReview forum: "TCBench: A Benchmark for Tropical Cyclone Track and Intensity Forecasting at the Global Scale"
_ICLR.cc/2026/Conference — Submitted to ICLR 2026_

### Official Review · Reviewer_WiLF · 2025-10-22

**Soundness:** 3
**Presentation:** 3
**Contribution:** 3
**Rating:** 4
**Confidence:** 4

**Summary:**

This paper introduces TCBench, a comprehensive benchmark designed for tropical cyclone forecasting. It provides standardized datasets and evaluation protocols to facilitate fair comparison and advancement in cyclone prediction research. The benchmark emphasizes reproducibility and realistic forecasting scenarios to encourage the development of more accurate and reliable models.

**Strengths:**

- This study presents a variety of benchmark tasks and experimental protocols for tropical cyclone prediction, encompassing data preprocessing, visualization tools, and evaluation metrics. In particular, it highlights the challenge of Rapid Intensification, pointing out the limitations of existing data-driven approaches in effectively capturing this phenomenon.

- In contrast to previous data-driven methods that have mainly focused on reducing errors in track prediction, this study emphasizes the importance of accurately forecasting intensity, which directly determines the potential damage tropical cyclones can cause to urban areas.

**Weaknesses:**

- In Line 144, the term “real-time-available data” is used, but ERA5 is a reanalysis dataset, which means it is not available in real time. Therefore, it seems that real-time prediction would not be possible through TCBench.

- This study proposes a benchmark framework that introduces various tasks and conducts experiments using baseline models. Since it aims to cover a wide range of aspects, there is still room for further experiments to demonstrate the utility of the benchmark. As shown in Figure 2(b), tropical cyclones of varying intensities occurred across different regions and seasons over six years, yet there is no detailed analysis of these cases. This work should include experiments analyzing the performance of each model across different seasons and scenarios, identifying where and when each model performs well or poorly.

**Questions:**

- As stated in Line 032 — “Tropical cyclones (TCs), also called ‘hurricanes’ or ‘typhoons’ depending on the basin” — the terminology differs depending on the region of occurrence. However, there appear to be no experiments analyzing the performance differences or prediction characteristics among these different types of tropical cyclones. Such an analysis could enhance the reliability and comprehensiveness of the benchmark dataset.
- Models such as typhoon trajectory prediction models presented in [1] and [2] could also serve as valuable baselines for comparison. Please provide the experimental results comparing with these models.
- The prediction of Rapid Intensification (RI) is formulated as a binary classification task, but it would be helpful to explain how this formulation contributes to practical disaster prevention and emergency response, particularly from a humanitarian or operational preparedness perspective.
- In Figure 4(c) of [3] (Pangu-Weather), the lead time of 120 hours corresponds to an error of approximately 200 km, whereas Figure 3(a) of TCBench shows that the track error of Pangu-Weather is significantly higher. Is this discrepancy due to different evaluation metrics between DPE and [3], or because different test datasets were used? Please provide the experimental results using consistent evaluation metrics and test datasets with those in [3].

[1] Huang, Cheng, et al. "MGTCF: multi-generator tropical cyclone forecasting with heterogeneous meteorological data." Proceedings of the AAAI Conference on Artificial Intelligence. Vol. 37. No. 4. 2023.

[2] Park, Young-Jae, et al. "Long-Term Typhoon Trajectory Prediction: A Physics-Conditioned Approach Without Reanalysis Data." The Twelfth International Conference on Learning Representations. 2023.

[3] Bi, Kaifeng, et al. "Accurate medium-range global weather forecasting with 3D neural networks." Nature 619.7970 (2023): 533-538.

**Details Of Ethics Concerns:**

Since this study proposes a dataset related to meteorology and climate, there are no relevant ethical concerns.

---

> ### Author Response · Authors · 2025-11-20
>
> Thank you for your thoughtful review of our work. We would like to further thank you for acknowledging our work and how it points out the limitations of current data-driven weather prediction in accurately forecasting intensity and rapid intensification. With regards to the weaknesses of our work, we would like to address these as follows:
>
> W1) We agree with the comment that if TCBench relies only on reanalysis data, it would not be able to be used operationally. However, we note that we only use reanalysis as the inputs for PanguWeather and FourCastNet v2, as the versions of these models available to us were not fine-tuned on operational analysis. We are happy to report, however, that we use operational analysis where applicable and possible (e.g., we use the record of HRES initial conditions made available by the ECMWF through the MARS client as inputs to the AIFS-Single 1.0 model). We note that the post-processing models thus use only ERA5 based initial conditions as Gomez et al. (2025) do not provide models trained on operational analysis. We have further clarified that the operational archive of HRES initial conditions is a data source in Table S1.
>
> W2) Thank you for pointing out the lack of analysis per year/season - this is important given that we limit the test set to 2023. This is because we expect that AIWP models will be trained on as much data as is available and the models we analyze are not trained/validated on data from 2023. However, we're happy to report that the routines we prepare are flexible and can be run on any number of additional years, and plan on encouraging the community to rely on them when conducting such analyses. This will be critical as we collect more data and can evaluate the models fairly (i.e., on unseen data) on longer timescales.
>
>
> With regards to your listed questions, we would like to address them as follows:
>
> Q1) Thank you for highlighting this issue - we agree that a per basin decomposition is important given differing behaviors and potential biases models can exhibit. We've implemented the evaluation routines in ways that will allow us to produce results for each basin, and are working on including a per basin decomposition in the appendix given the document length constraints.
>
> Q2) We thank you for suggesting additional references for inclusion in our work. Unfortunately, MGTCF model and "Long-Term Typhoon Trajectory Prediction" model provide data through 2019, and we focus on evaluating the performance of the models for the year 2023. However, we've added the references and discuss how these per-basin models are important in the introduction.
>
> Q3) We thank you for your feedback regarding how we introduce rapid intensification. We fully agree that the humanitarian implications are important and have added additional text and references in the text that introduces RI.
>
> Q4) We thank you for your feedback regarding the performance of PanguWeather as reported by TCBench and how it appears to be inconsistent with the original paper by Bi et al. (2023). We'd like to highlight that we include plots where we consider only those storms captured by each of the models (Fig. 6 in the appendix) and in these plots we find results that are consistent with the values in the original paper, which we believe highlights that when these models are able to predict a track for the storms they perform well - but that there is still a gap to be bridged considering the number of storms that are not captured in the model outputs.
>
> We thus note that the apparent inconsistency takes place when we consider all storms having taken place in 2023, and not just those storms that can be observed in the outputs of the model. Given the necessity of having some sort of fallback model for the storms where the neural weather models fail, we rely on persistence (which is an easily understood fallback model that admittedly performs quite poorly at extended lead times).
>
> We once again thank you for your valuable observations, and hope that our reply and revisions have addressed the weaknesses and questions regarding our work.

---

> ### Author Response · Authors · 2025-11-27
> **Follow-up**
>
> Dear reviewer WiLF,
>
> We wanted to follow up on your review and wanted to ask if you consider we've answered your questions on the basin specific evaluations and baseline models, as well as rapid intensification and the representation of PanguWeather on our results.
>
> If so, would you be willing to reconsider your score? Thank you for your time taken in reading this.

---

### Official Review · Reviewer_hG2W · 2025-10-31

**Soundness:** 2
**Presentation:** 2
**Contribution:** 2
**Rating:** 4
**Confidence:** 4

**Summary:**

TCBench​ is a benchmark for evaluating 1–5 day global tropical cyclone (TC) track and intensity forecasts. It uses ​IBTrACS​ observations and treats forecasting as predicting storm evolution from given initial conditions, covering both physical (TIGGE) and neural models (Pangu-Weather, etc.). While neural models perform well on track prediction, intensity forecasts need post-processing. Designed for accessibility, it provides evaluation metrics and tools to help AI and meteorology researchers improve TC prediction and understanding.

**Strengths:**

1.TCBench establishes a standardized and relatively fair evaluation pipeline. It uses IBTrACS as the "ground truth," converting all data into a unified format based on IBTrACS identifiers for consistency. For models that do not provide readily available tracks, it employs the unified TempestExtremes library with consistent parameters to derive tracks from raw model outputs. When a model fails to forecast a storm, TCBench does not simply ignore the sample but fills it using the persistence baseline to prevent models from selectively forecasting only easier-to-predict TCs.

2.TCBench features a comprehensive and multi-dimensional evaluation framework. It assesses both deterministic (DPE, RMSE, MAE) and probabilistic (CRPS) aspects of model predictions, providing a holistic view of model accuracy and reliability. It specifically designs evaluation metrics like CSI and PSS for Rapid Intensification (RI) events, addressing the gap of systematic RI evaluation in traditional assessments.

3.TCBench implements strict data inclusion criteria and a clear data split. Storms are partitioned by year into training (2017-2020), validation (2021-2022), and test (2023) sets, preventing data leakage and ensuring the consistency and validity of the evaluation.

4.TCBench integrates diverse data sources, provides multiple baseline models, and offers full toolchain support. It incorporates IBTrACS, ERA5, physics-based models (e.g., GEFS, IFS), and neural weather models (e.g., FourCastNetv2, Pangu-Weather, AIFS, GenCast). It provides various baseline models (e.g., MT-LB, Pangu, FourCastNet) and post-processing models (MLR, ANN, UNet). It also offers detailed experimental guidelines, with open-sourced code for reproducibility, and is designed for extensibility.

5.TCBench demonstrates strong experimental results and broad application potential. Regarding track forecasting, all models show practical utility, with some neural models exhibiting leading deterministic performance. For intensity forecasting, errors are smaller at longer lead times; post-processing mitigates the weakness of raw neural models, while physics-based models show robust baseline performance. Notably, only the post-processed models demonstrate any skill in forecasting RI. Furthermore, TCBench shows application potential in predicting key scalars for wind field reconstruction, nowcasting short-term precipitation and TC winds, developing wind downscaling models, predicting global TC activity across timescales, and identifying key environmental predictors for TC intensity.

**Weaknesses:**

1.TCBench relies on the IBTrACS observational dataset as the ground truth. While IBTrACS is the most complete and authoritative global TC archive currently available, it has limitations:
(1) Its quality varies by basin (e.g., lower reliability in the South Indian Ocean), inconsistencies exist in the initial track points determined by different agencies, and it lacks rigorous cross-validation against other data sources (e.g., regional satellite observations, ground radar data). Therefore, its absolute accuracy cannot be guaranteed, and using it as the definitive benchmark represents an idealization.
(2) For capturing RI events, the evaluation is confined to 24-hour windows, while IBTrACS data is at 6-hour intervals. Consequently, the precise timing and peak intensity of RI events might not be fully resolved, posing challenges for evaluating RI forecasts. As indicated by the experimental results, the assessment of extreme event forecasting capability remains insufficient.

2.When model-specific TC track information is unavailable, TCBench uses the TempestExtremes library to derive tracks from raw data. TempestExtremes identifies all candidate systems meeting its criteria, including short-lived, non-TC pressure systems. These must then be filtered by spatiotemporally matching them to known TCs in IBTrACS. This process introduces several issues:
(1) The algorithm detects numerous short-lived features (spurious tracks), increasing computational complexity and overhead.
(2) Applying the same unified TempestExtremes methodology to all model outputs may not optimally capture the TC center for each individual model.
(3) The matching process between TempestExtremes output and IBTrACS is complicated by significant disparities in model "coverage" (Figure 7) of the IBTrACS records. A realistically forecasted TC track that differs slightly from the IBTrACS record might be incorrectly filtered out as a "spurious track," thereby inflating the perceived model error.

3.In TCBench, using raw AI model outputs directly for TC intensity prediction often yields poor results (as shown in Figure 3). Consequently, post-processing models are introduced to refine the intensity forecasts from the raw neural weather models. This process itself has potential drawbacks:
(1) It may accumulate and amplify errors from the upstream AI model.
(2) As a purely data-driven optimization method, it can potentially produce results that are physically inconsistent from a meteorological perspective.
(3) The post-processing focuses solely on correcting intensity forecasts and does not address errors in track predictions.

**Questions:**

(1) Incorporate ablation experiments. The paper proposes numerous norms and innovations for improving the TC forecasting evaluation system, such as the unified TempestExtremes library for track derivation, the dedicated evaluation framework for Rapid Intensification (RI) events, and the strict partitioning of training, validation, and test sets by year. Conducting ablation studies using these specific features as model variants would effectively demonstrate the performance differences under various configurations. This would more robustly validate the critical importance of these design choices within the TCBench framework.

(2) In Section 3.2 (Data Inclusion Criteria), seven requirements are listed. However, the definitions of some criteria are overly vague. For instance:
- Criterion (e) does not specify which variables are included under "environmental fields".
- Criterion (g) lacks a precise definition and methodology for calculating "resolution" (which may vary across datasets).
It is recommended to add a "Detailed Specification Table for Data Inclusion Criteria" in an appendix to clarify these points explicitly.

(3) Integrate references to meteorological literature to strengthen analytical arguments and explanations. For example:
- When defining the forecasting tasks for TC track and intensity, cite relevant meteorological studies to elaborate on the driving factors of TC motion and the physical mechanisms governing intensity change.
- When analyzing RI prediction results, introduce literature discussing the inherent complexity and challenges in forecasting RI due to its multi-faceted causal factors.

(4) Provide a comprehensive workflow diagram illustrating the entire TCBench pipeline. A visual representation would greatly enhance the intuitive understanding of the process, from data ingestion and standardization to track processing, model evaluation, and benchmark generation.

---

> ### Author Response · Authors · 2025-11-20
>
> Thank you for your thoughtful review of our work. We appreciate your acknowledgment of our efforts towards developing a comprehensive and fair benchmark for tropical cyclone predictions by AI weather models and traditional numerical weather models. We would like to address each weakness and question individually, and have provided a preliminary revised manuscript with our changes marked in blue.
>
> With regards to the weaknesses, we address them as follows:
>
> W1: While it is true that IBTrACS has many limitations, especially when compared to basin-specific products, we rely on it as it is the best available record at the global scale, specifying that we choose the values reported by US agencies in order to avoid the inconsistent definitions of tropical cyclone intensity.
>
> With regards to the definition of rapid intensification (RI), we would like to highlight that while the definition we use is for 24-hour intensification, it is applied as a rolling window---meaning that we look at as many timesteps as possible while remaining consistent with the rapid intensification definition commonly used in meteorological literature. While we agree with there being limitations regarding our evaluation of RI (e.g., we are strict with timing consistency with observations), we also emphasize that the tracks derived from AI weather prediction models largely fail to predict *any* occurrences of RI, which further justifies your observations.
>
> W2: We thank you for the opportunity to discuss the limitations regarding our use of TempestExtremes, as we find that tracking is generally lightly discussed in the evaluation descriptions of the AIWP papers that introduce the models. As mentioned in aspect (2) of this weakness, we do not optimize the tracking algorithm for each one of the models. We do this for two reasons:
>
> a) We want to ensure that we use a consistent, well-known tracking algorithm with a proven record when we calculate tracks for models that do not directly provide them. We further note that there are efforts by the developers of these AIWP models to do the same, given the evolution of the TC tracking evaluation scheme used by DeepMind for their GraphCast article versus their more recent GenCast article; and
>
> b) While we rely on TempestExtremes for deriving said tracks, we have set up the evaluation framework so that users can themselves produce the tracks (including without TempestExtremes) and rely on TCBench only for the metric calculations and plotting. This provides the flexibility required to evaluate different prediction approaches, as models are often trained to specialize in single tasks (e.g., models that specifically predict RI; models that predict intensity and tracks directly from satellite observations).
>
> We also thank you for pointing out the computational overhead of TempestExtremes, as it's true that spurious tracks may increase the computations required. However, we found that the overhead is quite small compared to the compute required to run the AIWP models, and find it quite manageable even with single cores. We ascribe this to TempestExtremes being a non-data driven model, and fully agree that this would be a significant limitation if the tracking algorithm were to be replaced with a more computationally heavy alternative.
>
> Finally, we thank you for commenting on potential limitations regarding the matching process. We agree that any matching approach has the potential to eliminate tracks that are realistic but different from IBTrACS. However, we emphasize that we frame the problem as one focused on the use of AIWP for weather forecasting for existing tropical systems that are intense enough for agencies to keep a record of them. This, combined with our reliance on HuracanPy for matching predictions to observed tracks, serves to filter out tracks that are too far away from the observed reality (i.e., tracks whose initial-time positions are further than 300km away from the observed position at initial-time - further noting that initial-time is the time at forecast and not the time of cyclogenesis). Though this is not ideal for evaluations at longer  timescales (e.g., seasonal prediction), we argue that it is justified based on our focus on medium-range weather forecasting (2-14 days).

---

> ### Author Response · Authors · 2025-11-20
>
> Next we would like to answer each of the questions you've brought up and how we have addressed these in the updated manuscript:
>
> Q1) Thank you for suggesting that we run ablation experiments; we agree that this is a critical step for evaluating machine learning frameworks. However, we lack the compute resources to run ablation experiments on the AIWP models themselves as these require tremendous resources to train/finetune (e.g., 64 A100 GPUs for 1 week for AIFS). We hope that providing the resources needed to evaluate TC representations in these models allows AIWP model developers to incorporate such ablation experiments in their training pipeline. Similarly, while it is possible to change the parameters of TempestExtremes and the fields used in tropical cyclone tracking and in light of the extensive literature around TC tracking, we believe it best to present results run with the established, default configuration.
>
> Q2) Thank you very much for pointing out the vagueness of the terms "environmental fields" and "resolution" as present in the current text. In order to address this, we have added explanatory text and point to a more elaborate description in the supplementary materials per your recommendation, including a table listing the environmental fields.
>
> Q3) Thank you for this suggestion; we agree that adding references to the relevant meteorological literature would provide important context to the manuscript and can better guide readers familiar with TC physics. We've updated the introduction to address this feedback.
>
> Q4) Thank you for your feedback; we've updated Figure 1 to make the workflow clearer.
>
> Thank you again for your feedback, and we hope that our reply and revisions have addressed the weaknesses and questions regarding our work.

---

> ### Author Response · Authors · 2025-11-27
> **Follow-up**
>
> Dear reviewer hG2W,
>
> We wanted to follow up on your review and ask if you consider we've answered your questions regarding ablation experiments, data inclusion criteria, and additional references to meteorological literature.
>
> If so, would you be willing to reconsider your score? Thank you for your time taken in reading this.

---

### Official Review · Reviewer_13hc · 2025-11-01

**Soundness:** 3
**Presentation:** 3
**Contribution:** 3
**Rating:** 6
**Confidence:** 3

**Summary:**

The paper proposes TCBench, a benchmark for global tropical cyclone and intensity forecasting. TCBench uses IBTrACS observational data as the ground truth, and formulates TC forecasting as predicting the temporal evolution of both the location (latitude and longitude) and the intensity (maximum sustained wind speed). TCBench supports benchmarking both numerical models and data-driven methods like AIFS, Pangu-Weather, FourCastNetv2, and GenCast. TCBench provides deterministic and probabilistic storm-following metrics.

**Strengths:**

- The paper is original to the best of my knowledge.
- The paper is significant. It proposes a meaningful step forward for the field of data-driven weather forecasting, where most existing benchmarks evaluate the overall accuracy of methods while ignoring the equally important aspect of predicting extreme events like cyclones.
- The benchmark uses a standard data format and consistent evaluation pipelines, which ensures fairness and reproducibility.
- The benchmark provides post-processing steps to parse neural model outputs and supports both deterministic and probabilistic models.

**Weaknesses:**

- One major weakness is that the benchmark only considers forecasting tracks and the intensity of an existing cyclone, not an upcoming one. However, this is still a valid setting and has practical relevance.

**Questions:**

- How do we finetune a deep learning model with the provided cyclone data in TCBench?
- Is it possible to extend the benchmark to consider forecasting future/upcoming cyclones?

---

> ### Author Response · Authors · 2025-11-20
>
> Thank you for your thoughtful review and for commenting on the originality and importance of our work. We would like to address the identified weaknesses and questions as follows:
>
> We thank you for asking how TCBech can be used for finetuning deep learning models as we believe this is an important avenue for research. One way in which the tools provided with TCBench can be used is for deriving consistent definitions for TC intensity from IBTrACS without prior meteorological expertise. Providing consistently defined targets (i.e., avoiding the inconsistency in maximum wind definitions) should be helpful in the finetuning process. Additionally, certain metric definitions (e.g., fair CRPS for intensity, haversinial CRPS for tracks) may provide more TC-focused error quantification that can be combined to form a balanced loss for fine-tuning.
>
> We fully agree that predicting tropical cyclones before they form is a predictive avenue that holds much potential, especially considering that cyclogenesis is an active area of research. We decided to focus on existing tropical cyclones to provide comparisons of each model against observations, and thus allow a fair comparison between models based on how informative they could have been for forecasts made during the storms' lifetime.
>
> We contrast our approach with the approach taken in DeepMind's GraphCast article, where in order to compare the performance of GraphCast and HRES the analysis is done on the union of the predicted storms. This approach, which provides valuable insight into how the models represent cyclogenesis, has the unfortunate side effect of potentially reducing the size of the comparison set with each additional model that is considered. We do hope, however, to find ways of objectively and fairly evaluate the performance of the models on cyclogenesis in future iterations of our work.
>
> Thank you again for your feedback, and we hope that our reply and revisions have addressed the weaknesses and questions regarding our work.

---

> ### Author Response · Authors · 2025-11-27
> **Follow-up**
>
> Dear reviewer 13hc,
>
> We wanted to thank you again and follow up on your review - would you consider we've answered your questions regarding finetuning and extensions to the benchmark?
>
> If so, would you be willing to reconsider your score? Thank you for your time taken in reading this.

---

### Meta-Review · Area_Chair_MTRr · 2025-12-15

**Summary:**

This paper provides a benchmark for forecasting tropical cyclones with machine learning methods. Medium forecast lengths of up to five days are targeted, and several learned baselines are evaluated in parallel to a simulated one. Overall, tropical cyclones are certainly a challenging topic, but overall reviewers were not overly excited about the submission, stating concerns about the observational data, the postprocessing performed, and limited analysis.

Unfortunately, this paper only received three reviews, and no reviewers engaged prior to the cutoff time.

Given the lukewarm reception, I find it difficult to recommend acceptance despite a good chance for at least one possible raised scores. This is a borderline paper, and it seems to be positioned slightly below the bar. I can encourage the authors to expand their benchmark and its analysis in future resubmissions.

**Reviewer Concerns:**

Some complaints were very generic, such as suitability of observational data, and post-processing strategies. Naturally, these comments can't be addressed within the rebuttal period, and remain subject to individual assessments of priorities.

The authors also argue that further ablations are not reasonable given the computational requirements. They state that further analysis of the experiments is work in progress.

**Reviewer Scores:**

13hc already gave a 6 in a very short review

hG2W, with an initial score of 4, provided a detailed review outlining concerns about the observations, track information, and post-processing. Ablation experiments were asked for - the authors argued they are not reasonable. The other clarifications were addressed. It seems unlikely that the reviewer would have increased their score.

WiLF also gave a 4: the main weakness identified are further analysis of the experiments; the authors mention that experiments are in the works, but did not provide them in time.

---

### Decision · Program_Chairs · 2026-01-26

Reject